# Stress symptoms and positive coping during coronavirus disease 2019: The need to look at health from a gender perspective

Tania Romo-González[1]*, Antonia Barranca-Enríquez[2]*, Rosalba León-Díaz[1], José Manuel Hurtado-Capetillo[2], Socorro Herrera-Meza[3], Juan de Dios Villegas[4], Alejandro Botero Carvajal[5]

1 Biology and Integral Health Area of the Biological Research Institute, Universidad Veracruzana, Xalapa, Veracruz, México, 2 Center for Health Studies and Services, Universidad Veracruzana, Veracruz, México, 3 Psychological Research Institute, Universidad Veracruzana, Xalapa, Veracruz, México, 4 Medicine Program, Fundación Universitaria San Martin, Cali, Colombia, 5 Faculty of Health, Psychology Program, Universidad Santiago de Cali, Cali, Colombia

☯ These authors contributed equally to this work.
* tromogonzalez@uv.mx (TRG); abarranca@uv.mx (ABE)

**Data Availability Statement:** All relevant data are within the manuscript.

## Abstract

### Background

The emergency derived from coronavirus disease 2019 (COVID-19) has taught us important lessons in public and environmental health, particularly in the alarming numbers of existing noncommunicable diseases. However, one aspect to which little attention has been paid during the pandemic is mental health and its relationship with the gender perspective, in spite of gender being a determinant associated with health. In contrast, regarding health, few schemes and theories consider health from a positive and comprehensive perspective.

### Methods

This study was designed to examine the symptoms of stress and positive coping from a gender perspective. For this, the Stress Symptomatology Inventory, the Positive Coping to Life Scale and a general data questionnaire were applied to 665 individuals underwent the severe acute respiratory syndrome coronavirus 2 test at the Center for Health Studies and Services of the Universidad Veracruzana from July 2020 to November 2021.

### Findings

We found that women presented more stress symptoms and less positive coping in the factor of positive self-regulation of adverse situations and the factors of self-determination and positive self-regulation of important situations. Moreover, significant differences in the associations of these variables were observed between men and women.

**Funding:** This research was funded by the Dirección General de Investigaciones of Universidad Santiago de Cali, as part of the project "Fortalecimiento del Grupo GINEYSA" with code 450-621122-3508, in the form of a grant to ABC [02-2023].

**Competing interests:** The authors have declared that no competing interests exist.

## Conclusions

Therefore, the needs of women must be considered in the approach to the emergency department due to COVID-19 and in general in the health–disease process; therefore, not considering a gender approach will continue to deepen inequalities between sexes.

## Introduction

The pandemic caused by the severe acute respiratory syndrome coronavirus 2 (SARS-CoV-2) has affected almost all individuals directly or indirectly, either due to fear of becoming ill with coronavirus disease 2019 (COVID-19), due to the effects of far-reaching measures, or due to its economic and social impact [1]. Therefore, the pandemic has drastically virtually affected every aspect of our lives, leading to the biggest global recession since the Great Depression, and extreme social isolation because of changes in educational and work activities, closures of business premises, and restrictions on international travels [2]. All these have caused changes in the daily routines of individuals, limitations in social interactions, tensions between families locked together, and fear of getting sick and/or spreading the virus [1]. All this social isolation and financial instability, along with the fear of contracting COVID-19 and the uncertainty of the future, pose substantial physical and psychological stressors for the general population [2,3].

Although the COVID-19 situation has left us important lessons in terms of public and environmental health, particularly the alarming numbers of existing noncommunicable diseases, an aspect to which little attention has been paid during the pandemic is the relationship of the disease with the gender perspective and its consequences in the current context. Gender a determinant associated with health; however, when analyzing the differential consequences of the pandemic, it is observed that the gender perspective does not appear with the same eloquence in examining the direct and indirect effects of the pandemic compared with when addressing different fields of study.

Pandemics and outbreaks have differential effects on women and men, ranging from the risk of exposure and biological susceptibility to infection to social and economic consequences [4]. Additionally, as the capacities of health systems are overwhelmed, governments and facilities make decisions about prioritizing the provision of some health services and reducing others, such as sexual and reproductive health services, including pregnancy care, contraceptive provision, services for victims of sexual assault, and safe abortions, which may increase the risk of maternal mortality, unintended pregnancy and other adverse sexual and reproductive health outcomes for women and girls [4].

In contrast, women constitute 70% of the global health personnel and are highly represented in the front lines of the response. Note that violent attacks and harassment against health personnel in their homes and means of transportation took specific forms against women and generated differentiated impacts [5]. Furthermore, women also perform most of the unpaid care work, particularly home healthcare—the additional care burden brought on by COVID-19 for women must be recognized [4]. Thus, the COVID-19 pandemic has underscored society's reliance on women, both on the front lines and at home [6]. Additionally, some reports from several countries have indicated that when stay-at-home measures are put in place, the incidence of intimate partner and domestic violence increases. At the same time, during the COVID-19 pandemic, numerous reports were made that indicated increased levels of anxiety, stress, and a greater psychological impact on women [7,8].

For all of the above, global and national strategic response plans must be based on a robust gender analysis and ensure meaningful participation of affected groups, including women and girls, in decision-making and the implementation of programs.

Gender perspective in the field of health must be understood to eliminate those unnecessary, avoidable, and unfair disparities between women and men that are associated with systematic disadvantages in the socioeconomic context, to achieve comparable levels of physical, psychological and social well-being. All this implies that resources are allocated according to the specific needs of women and men, which services are received according to the particular needs of each gender, and that financing and payment for services are adjusted, to the economic capacity and not to the risks inherent to each gender or their needs. Therefore, adopting a gender perspective in the field of health implies linking the gender division of labor and power with the epidemiological profiles of a population and with the characteristics of accessibility, financing, and management of the health system [9].

Furthermore, note that some preconceived ideas lead to committing gender biases, including, the assumption that the health situation of women and men are the same, when in reality, they are not, or that there are differences by gender when there are similarities [10]. In this regard, the gender perspective proposes and allows for an in-depth analysis of the social relations between men and women, to clarify the differences and inequities in health associated with gender [10].

In contrast, concerning to health, this same patriarchal epistemology has generated few schemes and theories that consider positive integral health and well-being, because the predominant paradigm continues to be the medical–biologist that focuses on the study and treatment of the disease. In fact, when it comes to the study of stress, only recent scientific research in terms of psychological stress, coping with it, and health has changed the way health and coping skills can be understood [11–13]. In this sense, Góngora-Coronado [2010] proposed the concept of positive coping with life, implying the cognitive, emotional, and behavioral efforts that the person develops to handle external and/or internal demands in adverse or important situations of their life, taking advantage of their resources in the best possible way, to promote a fuller and happier life. Therefore, positive coping with life would occur not only in situations of stress or adversity, as in the original definition of this concept (coping) [14–16], but also in the face of different life situations that can represent an effort, not only to achieve an adaptation but also to promote a fuller life [17].

With all of the above, this study was designed to examine which aspects of positive coping are linked to the stress response, and with the presence/absence of important metabolic diseases in Mexico and in the context of the COVID-19 pandemic, from a gender perspective, in individuals who attended the Center for Studies and Health Services of the Universidad Veracruzana to undergo a polymerase chain reaction (PCR) test for COVID-19.

## Methods

### Sample

This was a descriptive and correlational study in which we gathered a non-probabilistic sample of patients (male or female older than 18 years) who attended the Center for Health Studies and Services (CESS) of the Universidad Veracruzana to take the PCR test for SARS-CoV-2.

Following all sanitary measures indicated during the COVID-19 pandemic, participants were verbally informed about the objectives of the protocol; once they agreed to participate, a Google Forms link via WhatsApp was sent to them, to obtain informed consent via online, and answer the general data questionnaire and the psychological tests. Personal data was processed by the policy of the Institute for Transparency, Access to Public Information, and

Protection of Personal Data. Once registered, a Google Forms link was sent through What-sApp to answer a file with general data and psychological tests. This work was carried out following the Code of Ethics of the Declaration of Helsinki and approved by the Committee of Ethical Research of the CESS.

## Data collection instruments

**General data questionnaire.**   The questionnaire was prepared expressly to collect data on age, sex, height, weight and the presence or absence of diabetes mellitus or hypertension.

**Symptoms of Stress Inventory (SSI).**   Stress symptoms are defined as all physical, functional, psychological and social manifestations that result from a person's exposure to stressful situations. Benevides, Moreno, Garrosa, and González [18] designed the Burnout Inventory for Psychologists, one part of which was the Inventory of Stress Symptomatology (SSI), an instrument that evaluates the frequency of the stress symptoms that occur in daily life. It consists of 30 items scored using a 5-point Likert: 0, never, 1, rarely, 2, moderately, 3, frequently and 4, always. It evaluates three factors: psychological symptoms (questions 10, 12, 14, 22, 4, 2, 30, 3, 17, 1, 29, and 18); physical symptoms (questions 27, 19, 15, 21, 9, 13, 25, and 5); and social symptoms (questions 16, 26, 6, 8, 28, 11, 20, and 24). These factors were scored as follows: psychological symptoms 0–11, low, 12–19, medium, and >19, high; physical symptoms 0–5, low, 6–8, medium, and >8, high; socio-psychological symptomatology from 0 to 5, low, from 6 to 8, medium and >8, high; global symptoms 0–22, low, 22–33, medium and >33, high. For psychological, social, and physical symptoms, the alpha coefficients were 0.848, 0.837 and 0.720, respectively [18]. This instrument was validated in Mexico by Moreno-Jiménez, Meda-Lara, Morante-Benadero, Rodríguez-Muñoz, and Palomera-Chávez [19]; the results showed an internal consistency, evaluated using Cronbach's alpha, with values between 0.87 and 0.78.

**Positive Coping with Life Scale (PCWLS).**   Positive coping with life is understood as the cognitive, emotional, and behavioral efforts that a person develops to handle external and/or internal demands in the face of adverse or important situations in their lives, taking advantage of their resources in the best possible way, promote a fuller and happier life. This scale was designed and validated by Góngora-Coronado and Vázquez-Velázquez [20]. It consists of 76 items, scored on a pictographic Likert-type scale with five response options (from Always to Never). Likewise, it evaluates 13 factors in total, divided into two subscales (situations) that meet the appropriate validity and reliability criteria. The adverse situation subscale includes the following factors 1) support from significant others, 2) positive self-regulation, 3) optimistic self-determination, 4) resilient attitude, 5) analysis and reflection, 6) planning and execution, and 7) personal effectiveness; whereas the important and/or significant situation subscale includes the following factors 1) self-determination, 2) analysis and positive assessment, 3) support from significant others, 4) positive self-regulation, 5) affective ties, and 6) optimistic attitude. Both subscales had a correlation coefficient of 0.832.

## Data analysis

An Excel database was created, and the assumptions of normality were assessed using the Kolmogorov–Smirnov test and homogeneity using the Levene test, before performing a Pearson correlation analysis to identify associations between variables. All analyses were performed using the Sigma Plot 10 program and the R environment (R Core Team). Given that in this sample, the number of men and women was not homogeneous, whether there are differences between the proportions of the following variables—age, body mass index (BMI), DB, and

HTA—between sexes, was verified using the Z test (Formula 1).

$$Z = \frac{(p1 - p2)}{EDD} = \frac{p1 - p2}{\sqrt{\frac{p1(1 - p1)}{n1} + \frac{p2(1 - p1)}{n2}}} \tag{1}$$

p1 = proportion 1

n1 = sample 1

p2 = proportion 2

n2 = sample 2

EDD = standard error of the difference in proportions

Likewise, the scores obtained in the subscales of the instruments used (see Methodology section) were compared between women and men, by age range, using the statistical program Sigma Stat, with the Kruskall–Wallis test, and they were graphed in the R Environment (Fig 4).

Finally, a network analysis is performed, allowing for the identification of the structures that emerge from the networks, which are a set of nodes or points related by lines. A relevant aspect describes a structural analysis of the networks, how each node is related within its set, and how sets are related to each other; in addition to the degree of the node, it is determined by the number of connections that a node presents, how close or far are the sets from the nodes, the nodes that do not cluster, etc. [21,22]. The process for the realization of the networks was based on the raw scores of the tests, Pearson correlations of each group were made, and later they were transformed to binary 0 being where there was no correlation and 1 when there was a correlation, and with these binarized correlation bases, the networks were created in the R Environment [23].

## Results

### Description of the population

There was a record of 607 patients who attended the CESS to be tested for COVID-19, of whom 56.6% were women and 43.4% were men with an average age of 45 years. Generally, the percentages of individuals with type 2 diabetes mellitus (8.3%), hypertension (17.6%), overweight (37%), and obesity (27.7%) were below the national average [24,25].

When analyzing the risk factors for COVID-19 (age, BMI, diabetes mellitus, and hypertension) between women and men, only in the case of normal weight did women present a higher and statistically significant percentage than men (38.37% and 24.33%, respectively), and men have a higher and statistically significant percentage in the case of age in the range of 70–89 years (6.84% and 3.19%, respectively) and obesity (31.94% and 23.54%, respectively) (Table 1).

### Women have higher levels of distress and less positive coping with life in terms of self-regulation and self-determination factors than men

When comparing the levels of distress between women and men, women presented higher and statistically significant values in all scales of the stress symptoms inventory (Fig 1). Likewise, on the scale of positive coping with life, women presented lower and statistically significant values in factor 2 (positive self-regulation) of adverse situations and in factors 1 (self-determination) and 4 (positive self-regulation) of important situations (Fig 2).

**Table 1. Comparison of risk factors for COVID-19 between women and men.**

| | Female | | Male | | X² | Z | gl | p |
|---|---|---|---|---|---|---|---|---|
| **Age** | **(n)** | **%** | **(n)** | **%** | | | | |
| 09–19 | 23 | 6.68 | 20 | 7.6 | 0.19 | 0.44 | 1 | 0.66 |
| 20–29 | 44 | 12.79 | 30 | 11.4 | 0.27 | 0.52 | 1 | 0.61 |
| 30–39 | 57 | 16.56 | 41 | 15.59 | 0.11 | 0.33 | 1 | 0.75 |
| 40–49 | 84 | 24.41 | 66 | 25.09 | 0.04 | 0.19 | 1 | 0.85 |
| 50–59 | 75 | 21.8 | 45 | 17.11 | 2.07 | 1.44 | 1 | 0.15 |
| 60–69 | 50 | 14.53 | 43 | 16.34 | 0.38 | 0.62 | 1 | 0.54 |
| 70–89 | **11** | **3.19** | **18** | **6.84** | **4.36** | **2.09** | **1** | **0.04*** |
| **BMI** | | | | | | | | |
| Underweight | 12 | 3.49 | 9 | 3.42 | 0.00 | 0.04 | 1 | 0.96 |
| Normal | 132 | **38.37** | **64** | 24.33 | **13.43** | **3.67** | **1** | **0.00**** |
| Overweight | 119 | 34.59 | 106 | 40.3 | 2.08 | 1.44 | 1 | 0.15 |
| Obesity | **81** | **23.54** | **84** | **31.94** | **5.30** | **2.30** | **1** | **0.02*** |
| **DB** | | | | | | | | |
| No | 312 | 90.69 | 226 | 85.93 | 3.36 | 1.83 | 1 | 0.07 |
| Unknown | 10 | 2.9 | 10 | 3.8 | 0.37 | 0.61 | 1 | 0.54 |
| Yes | 22 | 6.39 | 27 | 10.26 | 3.01 | 1.73 | 1 | 0.08 |
| **HTA** | | | | | | | | |
| No | 278 | 80.81 | 204 | 77.56 | 0.96 | 0.98 | 1 | 0.33 |
| Unknown | 9 | 2.61 | 10 | 3.8 | 0.69 | 0.83 | 1 | 0.41 |
| Yes | 57 | 16.56 | 49 | 18.63 | 0.44 | 0.66 | 1 | 0.51 |

## Older women and those with illnesses learn and face life better

When analyzing the associations between variables, women, unlike men, present positive correlations between age, BMI, and the presence of diabetes mellitus with positive coping subscales, and a negative correlation of age with psychic stress symptoms (Tables 2 and 3).

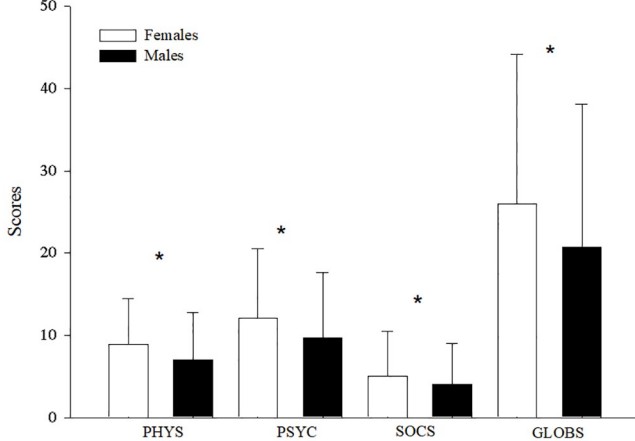

**Fig 1. Comparison of means and standard errors of stress symptoms between women and men.** PHYS: Physical stress symptoms; PSYC: Psychological stress symptoms; SOCS: Social stress symptoms; GLOBS: General symptoms of stress. *P<0.05.

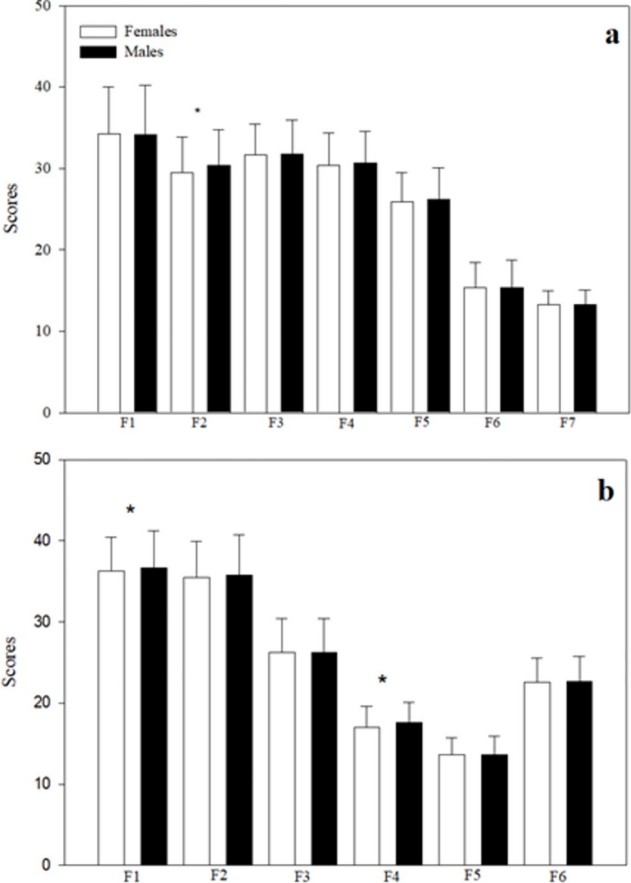

**Fig 2. Comparison of the means and standard errors of positive coping with stress between women and men.** a)
Adverse or difficult situations. F1: Support from significant people; F2: Positive self-regulation; F3: Optimistic self-
determination; F4: Resilient attitude; F5: Analysis and reflection; F6: Planning and execution; F7: Personal
effectiveness. b) Important or decisive situations. F1: Personal effectiveness; F2: Analysis and personal assessment; F3:
Search for support from significant others; F4: Positive self-regulation; F5: Affective ties; F6: Optimistic attitude.
*P<0.05.

In contrast, men presented associations between physical stress symptoms and arterial
hypertension, between social stress symptoms and diabetes mellitus and hypertension, and
between general stress symptoms and hypertension.

Finally, both women and men presented negative correlations between stress symptoms
and positive coping with stress (Tables 2 and 3).

Note that when analyzing the spatial distribution and the degree of centrality of the correla-
tions of the variables between women and men, they varied according to the pathology pre-
sented (Fig 3). For example, in the case of hypertension and more markedly diabetes mellitus,
both women and men lose connectivity networks with the presence of the diseases. However,
in women, although some nodes have less connectivity than those in men, the degree of cohe-
sion of the network (closeness and betweenness) with the disease was greater, in addition to
the rearrangement of the nodes in which the variables of stress symptoms and positive coping
are compartmentalized (Fig 3).

When the data were analyzed by age and gender, we observed significant differences in the
presentation of stress symptoms between women and men, particularly those in middle ages

**Table 2. Correlations between risk factors for COVID-19, stress symptoms, and positive coping in women.**

| | BMI | Db | HBP | PHYS | PSYC | SOCS | GLOBS | F1 Adv | F2 Adv | F3 Adv | F4 Adv | F5 Adv | F6 Adv | F7 Adv | F1 Imp | F2 Imp | F3 Imp | F4 IMP | F5 IMP | F6 IMP |
|---|---|---|---|---|---|---|---|---|---|---|---|---|---|---|---|---|---|---|---|---|
| Age | 0.34 | 0.27 | 0.43 | 0.00 | -0.11 | -0.01 | -0.05 | 0.07 | 0.15 | 0.20 | 0.10 | 0.12 | 0.11 | 0.11 | 0.17 | 0.10 | 0.10 | 0.12 | 0.15 | 0.19 |
| BMI | | 0.21 | 0.33 | 0.06 | 0.03 | -0.02 | 0.03 | 0.12 | 0.17 | 0.20 | 0.14 | 0.18 | 0.04 | 0.10 | 0.20 | 0.18 | 0.10 | 0.10 | 0.15 | 0.18 |
| Db | | | 0.28 | 0.00 | -0.08 | -0.04 | -0.05 | 0.05 | 0.05 | 0.07 | 0.04 | 0.07 | -0.01 | 0.04 | 0.11 | 0.11 | 0.05 | 0.04 | 0.11 | 0.10 |
| HBP | | | | 0.13 | 0.01 | 0.05 | 0.06 | 0.04 | 0.06 | 0.10 | 0.03 | 0.09 | -0.01 | 0.02 | 0.08 | 0.06 | 0.01 | 0.02 | 0.09 | 0.08 |
| PHYS | | | | | 0.75 | 0.72 | 0.89 | -0.36 | -0.40 | -0.30 | -0.40 | -0.35 | -0.32 | -0.33 | -0.29 | -0.30 | -0.34 | -0.37 | -0.26 | -0.38 |
| PSYC | | | | | | 0.81 | 0.95 | -0.42 | -0.48 | -0.38 | -0.46 | -0.41 | -0.39 | -0.40 | -0.38 | -0.36 | -0.40 | -0.50 | -0.33 | -0.45 |
| SOCS | | | | | | | 0.90 | -0.43 | -0.47 | -0.39 | -0.46 | -0.42 | -0.34 | -0.43 | -0.38 | -0.39 | -0.41 | -0.44 | -0.34 | -0.45 |
| GLOBS | | | | | | | | -0.44 | -0.49 | -0.38 | -0.47 | -0.43 | -0.38 | -0.41 | -0.38 | -0.38 | -0.42 | -0.48 | -0.34 | -0.47 |
| F1 Adv | | | | | | | | | 0.71 | 0.66 | 0.70 | 0.78 | 0.60 | 0.63 | 0.61 | 0.66 | 0.83 | 0.64 | 0.74 | 0.66 |
| F2 Adv | | | | | | | | | | 0.81 | 0.80 | 0.83 | 0.61 | 0.76 | 0.79 | 0.79 | 0.72 | 0.84 | 0.61 | 0.77 |
| F3 Adv | | | | | | | | | | | 0.78 | 0.79 | 0.53 | 0.78 | 0.83 | 0.77 | 0.67 | 0.71 | 0.59 | 0.76 |
| F4 Adv | | | | | | | | | | | | 0.81 | 0.64 | 0.75 | 0.76 | 0.75 | 0.67 | 0.68 | 0.59 | 0.73 |
| F5 Adv | | | | | | | | | | | | | 0.65 | 0.74 | 0.76 | 0.82 | 0.75 | 0.70 | 0.61 | 0.70 |
| F6 Adv | | | | | | | | | | | | | | 0.53 | 0.54 | 0.62 | 0.58 | 0.56 | 0.46 | 0.48 |
| F7 Adv | | | | | | | | | | | | | | | 0.75 | 0.74 | 0.67 | 0.69 | 0.57 | 0.69 |
| F1 Imp | | | | | | | | | | | | | | | | 0.86 | 0.69 | 0.78 | 0.62 | 0.81 |
| F2 Imp | | | | | | | | | | | | | | | | | 0.75 | 0.77 | 0.61 | 0.76 |
| F3 Imp | | | | | | | | | | | | | | | | | | 0.70 | 0.73 | 0.72 |
| F4 Imp | | | | | | | | | | | | | | | | | | | 0.59 | 0.77 |
| F5 Imp | | | | | | | | | | | | | | | | | | | | 0.64 |

**Table 3. Correlations between risk factors for COVID-19, stress symptoms, and positive coping in men.**

| | BMI | Db | HBP | PHYS | PSYC | SOCS | GLOBS | F1 Adv | F2 Adv | F3 Adv | F4 Adv | F5 Adv | F6 Adv | F7 Adv | F1 Imp | F2 Imp | F3 Imp | F4 Imp | F5 Imp | F6 Imp |
|---|---|---|---|---|---|---|---|---|---|---|---|---|---|---|---|---|---|---|---|---|
| Age | 0.13 | 0.17 | 0.28 | 0.03 | 0.05 | 0.08 | 0.06 | -0.01 | 0.07 | 0.05 | 0.06 | 0.07 | 0.06 | 0.00 | 0.05 | 0.06 | 0.04 | 0.07 | 0.02 | 0.02 |
| BMI | | 0.14 | 0.17 | 0.04 | -0.09 | -0.09 | -0.05 | 0.01 | 0.08 | 0.15 | 0.13 | 0.11 | 0.01 | 0.12 | 0.17 | 0.17 | 0.11 | 0.13 | 0.09 | 0.15 |
| Db | | | 0.29 | 0.11 | 0.10 | 0.16 | 0.12 | -0.07 | 0.01 | -0.01 | 0.00 | 0.03 | -0.06 | 0.01 | -0.01 | 0.03 | -0.01 | -0.04 | 0.01 | -0.07 |
| HBP | | | | 0.18 | 0.11 | 0.14 | 0.15 | -0.07 | -0.06 | -0.01 | -0.04 | -0.02 | 0.01 | 0.00 | 0.02 | 0.00 | -0.06 | -0.10 | -0.08 | -0.10 |
| PHYS | | | | | 0.80 | 0.68 | 0.91 | -0.32 | -0.35 | -0.31 | -0.36 | -0.33 | -0.35 | -0.30 | -0.28 | -0.26 | -0.30 | -0.32 | -0.26 | -0.36 |
| PSYC | | | | | | 0.80 | 0.96 | -0.42 | -0.48 | -0.44 | -0.48 | -0.44 | -0.44 | -0.41 | -0.43 | -0.41 | -0.42 | -0.49 | -0.33 | -0.49 |
| SOCS | | | | | | | 0.86 | -0.38 | -0.42 | -0.41 | -0.43 | -0.41 | -0.42 | -0.39 | -0.41 | -0.37 | -0.37 | -0.40 | -0.32 | -0.47 |
| GLOBS | | | | | | | | -0.40 | -0.45 | -0.42 | -0.46 | -0.42 | -0.44 | -0.40 | -0.41 | -0.38 | -0.39 | -0.44 | -0.32 | -0.47 |
| F1 Adv | | | | | | | | | 0.69 | 0.66 | 0.70 | 0.76 | 0.68 | 0.65 | 0.61 | 0.64 | 0.80 | 0.59 | 0.70 | 0.64 |
| F2 Adv | | | | | | | | | | 0.82 | 0.79 | 0.82 | 0.70 | 0.72 | 0.76 | 0.80 | 0.69 | 0.82 | 0.52 | 0.74 |
| F3 Adv | | | | | | | | | | | 0.75 | 0.79 | 0.63 | 0.71 | 0.79 | 0.76 | 0.64 | 0.65 | 0.54 | 0.73 |
| F4 Adv | | | | | | | | | | | | 0.80 | 0.74 | 0.75 | 0.73 | 0.73 | 0.63 | 0.65 | 0.53 | 0.70 |
| F5 Adv | | | | | | | | | | | | | 0.74 | 0.80 | 0.75 | 0.83 | 0.71 | 0.66 | 0.55 | 0.66 |
| F6 Adv | | | | | | | | | | | | | | 0.61 | 0.59 | 0.65 | 0.62 | 0.57 | 0.42 | 0.56 |
| F7 Adv | | | | | | | | | | | | | | | 0.69 | 0.72 | 0.58 | 0.58 | 0.45 | 0.59 |
| F1 Imp | | | | | | | | | | | | | | | | 0.88 | 0.71 | 0.77 | 0.54 | 0.81 |
| F2 Imp | | | | | | | | | | | | | | | | | 0.74 | 0.76 | 0.55 | 0.75 |
| F3 Imp | | | | | | | | | | | | | | | | | | 0.70 | 0.74 | 0.72 |
| F4 Imp | | | | | | | | | | | | | | | | | | | 0.54 | 0.77 |
| F5 Imp | | | | | | | | | | | | | | | | | | | | 0.61 |

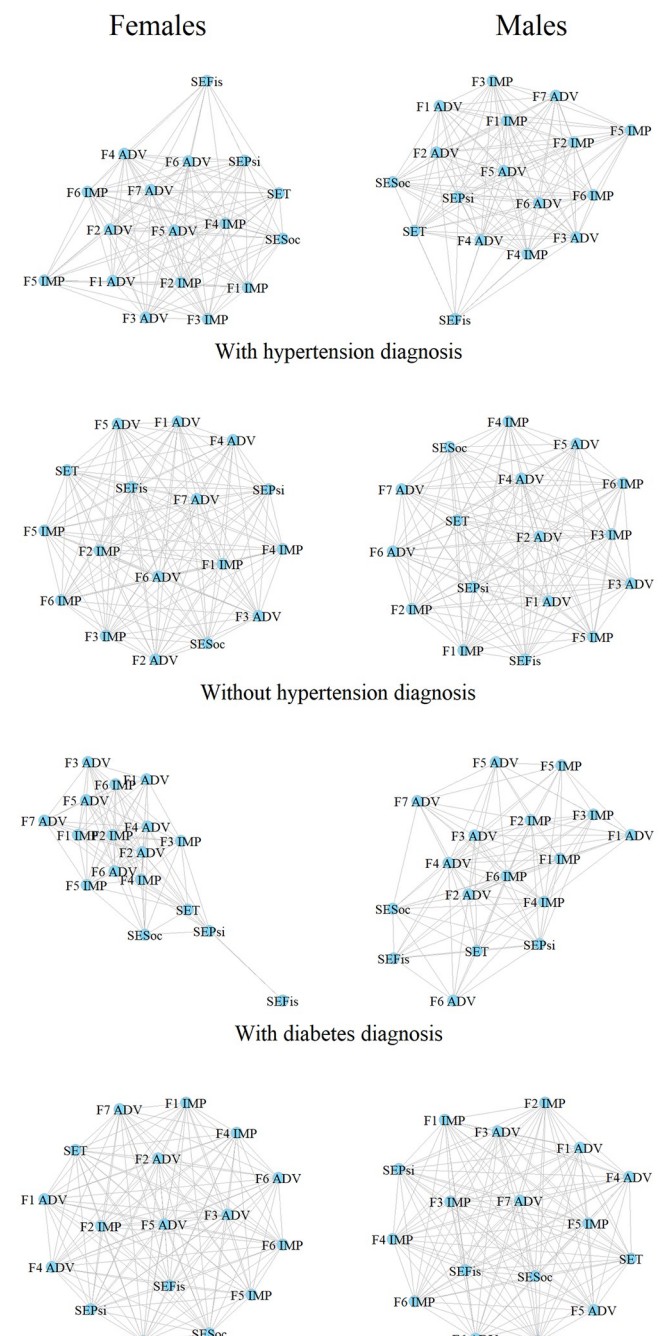

**Fig 3. Spatial and structural organization of the networks in women and men according to the pathology (with hypertension vs. without hypertension, and with diabetes vs. without diabetes) in women and men.** Adverse or difficult situations: ADV. F1: Support from significant people; F2: Positive self-regulation; F3: Optimistic self-determination; F4: Resilient attitude; F5: Analysis and reflection; F6: Planning and execution; F7: Personal effectiveness. Important or Decisive Situations: IMP. F1: Personal effectiveness; F2: Analysis and personal assessment; F3: Search for support from significant others; F4: Positive self-regulation; F5: Affective ties; F6: Optimistic attitude.

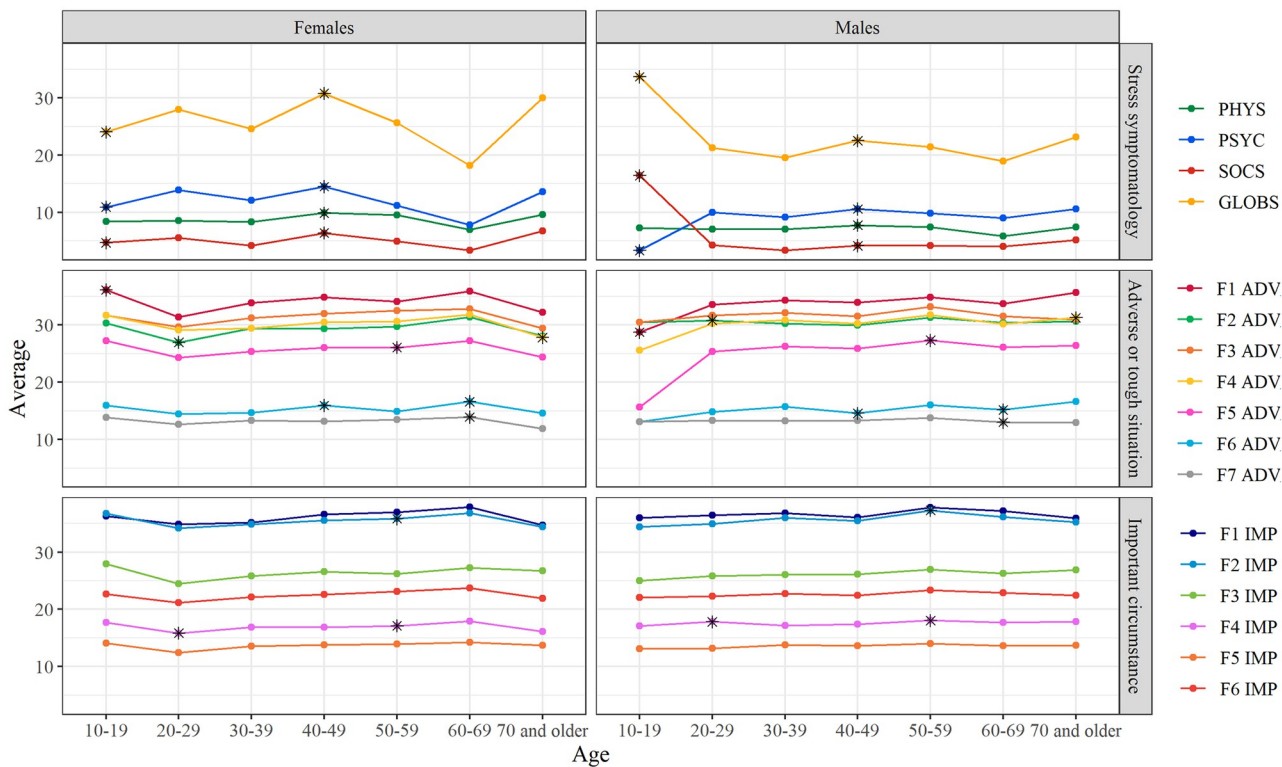

**Fig 4. Differences by sex and age in the scores obtained using the stress symptom inventory and the Positive Coping with Life Scale.** *P<0.05.

(40–49 years), when women have the greatest burden of family responsibilities. Likewise, stress scores decreased between the ages of 60 and 69 years, because family responsibilities decreased (Fig 4). Likewise, in women, there was a greater propensity to plan future actions in these same age ranges.

## Discussion

The pandemic has affected almost everyone directly or indirectly; thus, in addition to its physical effects, COVID-19 has caused significant psychological stress among those affected [1–3].

Although the emergency derived from COVID-19 has left us important lessons, one of the aspects to which little attention has been paid during the pandemic is its relationship with the gender perspective and its consequences in the current context, considering that pandemics and outbreaks have differential effects on women and men, ranging from the risk of exposure and biological sensitivity to infection to social and economic consequences [4].

In this regard, the data that we presented in this research showed differences in the physical risk factors for contracting COVID-19, because men presented statistically significant differences in the case of obesity and in the age range from 70 to 89 years. Therefore, men were more exposed to infection, as indicated by the literature, in addition to other comorbidities present in them [26]. Particularly in Mexico, hypertension (45.53%), diabetes mellitus (39.39%), and obesity (30.4%) were reported as the main comorbidities in deceased patients [27]. In the case of the state of Veracruz, diabetes occurs as the second most prevalent disease, and cardiovascular diseases are the second cause of death [27,28], and the city of Veracruz, where the data were collected, was the city with more Covid-19 reported cases.

Although we found high levels of stress associated with gender, there are inconsistent findings in the literature on the perception of stress linked to gender [29], because the specific reasons for reporting higher levels of stress in women have not yet been elucidated, whether they are health personnel or the common population [30]. Furthermore, most studies consider that being female is more related to the presence of stress symptoms, and psychological effects from the pandemic [7]. In our study, the main finding is the identification of positive coping factors that helped modulate the effect of a common stressor: COVID-19. Regarding this, when reviewing those psychological factors, women presented higher scores in stress symptoms and its subscales of physical, mental, and social symptoms—a phenomenon also observed by other investigations that showed that women tended to report higher stress and that women presented positive correlations between gender and stress [1–3,7]. These variations could be because women comprised 70% of the global health workforce and were highly represented in the front lines of the response, therefore, they were at a high risk of frequent exposure not only to patients with high virus loads but also to physical and emotional wear and tear; additionally, because of their gender role as caregivers during isolation at home, they performed most unpaid care tasks, having to combine work, home responsibilities, and childcare during isolation. Likewise, the incidence of domestic and partner violence increased [4,31]. Despite these differences in stress symptom scores, women presented values similar to those of men in positive coping with stress, indicating an important source of resilience for women. That is, most individuals can sufficiently cope with the pandemic and its associated measures [1]. Only in the case of the positive self-regulation factor of adverse situations and in the self-determination and positive self-regulation factors of important situations did women score lower than men, which again could be because of the greater demand that women had and emotional exhaustion, making it possible to control or manage emotions in a positive way, to remain calm and try to be at peace, and to analyze the situation. The practical sense and the security to solve it despite the obstacles were strongly affected in women. These processes are based on the neurobiological response that is activated in women of "help and protection", associated with the activation of the limbic system, which, although it favors care, also triggers the stress response [32].

In addition to the results in the variation of stress symptom scores and positive coping with stress between women and men, it is also interesting to note that the correlation analyses showed differences that could be related not only to the gender role played by women for generations but also to the evolutionary changes resulting from these roles. This is the case of the correlations presented by women with age and the various subscales of positive coping (Difficult Situations: Positive self-regulation, Optimistic self-determination, Resilient attitude, Analysis and reflection, Planning and execution, and Personal effectiveness; and Important or Decisive Situations: Personal effectiveness, Positive self-regulation, Affective ties and Optimistic attitude) and the BMI with the factors support of significant people, Positive self-regulation, Optimistic self-determination, Resilient attitude and Analysis and reflection of adverse or difficult situations and Personal effectiveness, Analysis and personal assessment, Affective bonds and optimistic attitude of Important or Decisive Situations; as well as the presence of diabetes with the factors Personal effectiveness and Analysis and personal assessment of Important or Decisive Situations. All this seems to indicate that women with age and illness learn to face life better.

In contrast, men, presenting only associations between physical stress symptoms and arterial hypertension, social stress symptoms and diabetes mellitus and hypertension and between general stress symptoms, seem to only react to diseases with more symptoms. This could be associated with the fact that the main coping strategies in men are more cognitive, which can lead them to either solve or ignore the problem [33].

This relationship of gender with the pathologies is much clearer when analyzing the spatial distribution and the degree of centrality of the correlations of the variables between women and men with the pathology presented, because the networks in the case of women (although they lose connectivity, they gain in cohesion), are much more dramatic in the case of diabetes mellitus (Fig 3 and Supplementary Material).

Finally, there were some age windows more important for women and others for men when it comes to stress symptoms and positive coping. In the case of women, significant differences in stress symptoms were observed among women in the age range of 40–49 years, which may be related to the pre-menopause phase [34]. In the case of men, some differences appeared regarding the symptoms of stress in the age range of 60–69 years. These differences could be associated with biological aspects (e.g., decreased testosterone release and increased progesterone release), and simultaneously may be associated with social aspects, such as retirement, the diagnosis of terminal illnesses, or aging itself, which could lead to states of stress and changes in life perspective, and as shown in the changes in the Personal Analysis and Appraisal, Resilient Attitude, and Analysis and Reflection scores [35]. Aging in particular has been proposed as an important source of stress, simultaneously implies the patent experience of physical and mental deterioration [36], which was mainly noticeable in men in this study.

In this sense, the incorporation of women's needs in the emergency approach is not a minor issue; in contrast, not considering a gender approach will deepen the inequalities with effects that will last in the long term and will be difficult to reverse. Therefore, this reality requires that the equal participation of women in decision-making and the gender approach are central elements of crisis mitigation and recovery policies.

Lastly, this article showed the analysis of the data collected during more than 12 months of the course of the pandemic, from people of the city of Veracruz who requested a PCR test for the diagnosis of SARS CoV-2. The city of Veracruz was one of the cities with the highest number of cases during the first months after the declaration of a pandemic by COVID-19 and it remained in red epidemiological light for longer than most of the municipalities in the state of Veracruz. It is important to point out that the city of Veracruz also had a greater number of hospitals with diagnostic capacity for SARS CoV-2, but given that the number of confirmed cases was always high, the population that had the possibility of performing the diagnostic test in another site, looked for alternatives, such as going to the CESS, hence our sampling was not random, but for convenience. Therefore, we cannot ensure the total representativeness of the population; however, the data was rigorously analyzed, and we sought to contextualize it appropriately for the state of the pandemic during sampling.

## Acknowledgments

Authors would like to thank Israel Huesca Rodríguez for his help with the design of the figures.

## Author Contributions

**Conceptualization:** Tania Romo-González, Antonia Barranca-Enríquez.

**Data curation:** Antonia Barranca-Enríquez.

**Formal analysis:** Tania Romo-González, Rosalba León-Díaz.

**Investigation:** Tania Romo-González, Antonia Barranca-Enríquez.

**Methodology:** Tania Romo-González, Rosalba León-Díaz.

**Project administration:** Antonia Barranca-Enríquez.

**Writing – original draft:** Tania Romo-González, Antonia Barranca-Enríquez.

**Writing – review & editing:** Tania Romo-González, Antonia Barranca-Enríquez, Rosalba León-Díaz, José Manuel Hurtado-Capetillo, Socorro Herrera-Meza, Juan de Dios Villegas, Alejandro Botero Carvajal.

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
