## [Decision Letter · Decision Letter 0]

16 Mar 2023

PONE-D-22-34111STRESS SYMPTOMS AND POSITIVE COPING DURING CORONAVIRUS DISEASE 2019: THE NEED TO LOOK AT HEALTH FROM A GENDER PERSPECTIVEPLOS ONE

Dear Dr. Romo-González,

Thank you for submitting your manuscript to PLOS ONE. After careful consideration, we feel that it has merit but does not fully meet PLOS ONE’s publication criteria as it currently stands. Therefore, we invite you to submit a revised version of the manuscript that addresses the points raised during the review process.

The manuscript presents a description of the methodology, the instruments used and the strategies for data collection, however, the recruitment of people who accepted a Covid-19 test is the only selection criteria.

The context of the pandemic in that locality during the study period and the conditions related to the infection that could support the conclusions, for example, severity, disability, are unknown. The conclusions must consider the factors related to the infection and not only the comorbidities and it would be interesting to see if the disease had an effect on coping.

The network analysis was done taking into account the scores of the subscales, however it is not clear how these aspects determine health and well-being with a gender perspective. It could be useful to do the network analysis with demographic variables and comorbidities

There is no description of biases or how to control them

In the introduction, the gender perspective is blurred from the pandemic and the specific situation related to acute respiratory infection by SARS-CoV-2, it is raised from health in general, which makes it difficult to understand the working hypothesis.

It remains to describe the municipality and the covid situation in the area at the project development site, which are contextual elements that can contribute to the analysis and generation of conclusions

The variables that are going to be used to analyze the gender perspective are not described, there is only sex. It is not clear if the PCWLS scale was validated in the sample studied.

Among the results, they affirm that differences were found in risk factors for contracting covid, it must be taken into account that this type of study does not allow us to rigorously identify these differences in risk factors, what is proposed is the analysis of the comorbidities of the people positive for Covid-19

It remains to be clarified whether it was possible to identify the analysis of the aspects that determine health and well-being from a gender perspective in the methods and present it in the conclusions in people with confirmed Covid-19

The selection bias that occurs when selecting people who consult a health service for symptoms of Covid-19 must be declared, since they may have different conditions or characteristics from those who do not consult.

Please take into account the reviewer's comments and consider the information provided to complement the conclusions

We look forward to receiving your revised manuscript.

Kind regards,

Diana Marcela Walteros Acero, M.D.

Academic Editor

PLOS ONE

Journal Requirements:

"This research has been funded by Dirección General de Investigaciones of Universidad Santiago de Cali under call No. 01-2022."

"This research has been funded by Dirección General de Investigaciones of Universidad Santiago de Cali under call No. 01-2022. The funders had no role in study design, data collection and analysis, decision to publish, or preparation of the manuscript."

Reviewers' comments:

Reviewer's Responses to Questions

**Comments to the Author**

1. Is the manuscript technically sound, and do the data support the conclusions?

Reviewer #1: Yes

2. Has the statistical analysis been performed appropriately and rigorously? 

Reviewer #1: Yes

3. Have the authors made all data underlying the findings in their manuscript fully available?

Reviewer #1: Yes

4. Is the manuscript presented in an intelligible fashion and written in standard English?

Reviewer #1: Yes

5. Review Comments to the Author

Reviewer #1: I recommend the authors to mention the following findings related to gender, COVID-19 and mental health in the Introduction and discussion, I will review this paper again.

Search PubMed for: Female gender, student status, specific physical symptoms (e.g., myalgia, dizziness, coryza), and poor self-rated health status were significantly associated with a greater psychological impact of the outbreak and higher levels of stress, anxiety, and depression (p < 0.05)

Search PubMed for: After adjusting for age, gender and comorbidities, it was found that depression (OR 2.79, 95% CI 1.54-5.07, p = 0.001), anxiety (OR 2.18, 95% CI 1.36-3.48, p = 0.001), stress (OR 3.06, 95% CI 1.27-7.41, p = 0.13), and PTSD (OR 2.20, 95% CI 1.12-4.35, p = 0.023) remained significantly associated with the presence of physical symptoms experienced in the preceding month.

Search PubMed for: Risk factors associated with distress measures include female gender, younger age group (≤40 years), presence of chronic/psychiatric illnesses, unemployment, student status, and frequent exposure to social media/news concerning COVID-19.

Search PubMed for: Female gender; youth age; single status; students; specific symptoms; recent imposed quarantine; prolonged home-stay; and reports of poor health status, unnecessary worry, concerns for family members, and discrimination were significantly associated with greater psychological impact of the pandemic and higher levels of stress, anxiety and depression (p<0.05).

Search PubMed for: Comprehensive strategies for the screen of psychological problems and to support high-risk groups are critical, especially females, middle-aged adults and the elderly, affected laborers, and health care professionals.

Search PubMed for: Being female, having chronic conditions and living in the family with 3-5 members were associated with lower HRQOL scores. A comprehensive assessment of the influence of COVID-19 along with public health interventions, especially mental health programs

Search PubMed for: Higher awareness of local pandemic situation was associated with female respondents (Coef.: 6.19; 95% CI: 0.51; 11.87) and larger family sizes of above 5 people (Coef.: 9.00; -1.00; 19.00). Respondents between 35-44 years old were shown to be less aware of preventive behavioral practices than other age groups, including the group of participants above 44 years old (Coef.: -0.34; 95% CI: -0.67; -0.02).

Search PubMed for: Seventy pregnant women (8.1%) reported that their antenatal care was influenced by the COVID-19. In this group, a higher level of satisfaction with the care of parents-in-law

Search PubMed for: Of 651 pregnant women, 60.4% accepted to receive the vaccine, and 82.6% of the total pregnant women were willing to pay for a COVID-19 vaccine with the mean amount of WTP of USD 15.2 (SD ± 27.4).

Search PubMed for: The subgroups identified to have a higher risk of psychiatric symptoms among the general public include females, the elderly, individuals with chronic illness, migrant workers, and students.

Please discuss the role of female gender based on the following papers.

Investigating Psychological Differences Between Nurses and Other Health Care Workers From the Asia-Pacific Region During the Early Phase of COVID-19: Machine Learning Approach. JMIR Nurs. 2022 Jun 1;5(1):e32647. doi: 10.2196/32647. PMID: 35648464; PMCID: PMC9162133.

6. PLOS authors have the option to publish the peer review history of their article (what does this mean?). If published, this will include your full peer review and any attached files.

Reviewer #1: No

---

## [Author Response · Author response to Decision Letter 0]

17 Apr 2023

Diana Marcela Walteros Acero, M.D.

Academic Editor

PLOS ONE

Regarding the manuscript entitled " Stress symptoms and positive coping during coronavirus disease 2019: the need to look at health from a gender perspective, that we submitted to the PLOS ONE, I take this opportunity to thank the reviewers for their comments and suggestions. 

Comments from the Editor: 

The context of the pandemic in that locality during the study period and the conditions related to the infection that could support the conclusions, for example, severity, disability, are unknown. The conclusions must consider the factors related to the infection and not only the comorbidities and it would be interesting to see if the disease had an effect on coping.

R= We explain the context of the epidemic in the state of Veracruz in the following text:

In the case of the state of Veracruz, diabetes occurs as the 2nd. most prevalent disease, and cardiovascular diseases are the second cause of death (27-28), and the city of Veracruz, were the data were collected, was the city with more Covid-19 reported cases. 

The city of Veracruz was one of the cities with the highest number of cases during the first months after the declaration of a pandemic by COVID-19 and it remained at a red epidemiological light for longer than most of the municipalities in the state of Veracruz. It is important to point out that the city of Veracruz also had a greater number of hospitals with diagnostic capacity for SARS CoV-2, but given that the number of confirmed cases was always high, the population that had the possibility of performing the diagnostic test in another site, looked for alternatives, such as going to the CESS, hence our sampling was not random, but for convenience.

2. The network analysis was done taking into account the scores of the subscales, however it I s not clear how these aspects determine health and well-being with a gender perspective. It could be useful to do the network analysis with demographic variables and comorbidities. There is no description of biases or how to control them.

R=Although the network analysis considers the factors that make up the two subscales of the positive coping instrument separately, the factors show more clearly how participants can cope with difficult situations, or fully enjoy positive situations.

3. In the introduction, the gender perspective is blurred from the pandemic and the specific situation related to acute respiratory infection by SARS-CoV-2, it is raised from health in general, which makes it difficult to understand the working hypothesis.

R=We have reduced the text that refers to the pandemic, so as not to divert attention from the gender perspective and how it should be considered in health.

4. It remains to describe the municipality and the covid situation in the area at the project development site, which are contextual elements that can contribute to the analysis and generation of conclusions. 

R=In the case of the state of Veracruz, diabetes occurs as the 2nd. most prevalent disease, and cardiovascular diseases are the second cause of death (27-28), and the city of Veracruz, where the data were collected, was the city with more Covid-19 reported cases. 

5. The variables that are going to be used to analyze the gender perspective are not described, there is only sex. It is not clear if the PCWLS scale was validated in the sample studied.

R= The PCWLS scale was validated by by Góngora-Coronado and Vázquez-Velázquez. 

6. Among the results, they affirm that differences were found in risk factors for contracting covid, it must be taken into account that this type of study does not allow us to rigorously identify these differences in risk factors, what is proposed is the analysis of the comorbidities of the people positive for Covid-19. 

R=According to the purpose of the study, we analyzed which aspects of positive coping are linked to the stress response, and with the presence/absence of important metabolic diseases in Mexico and in the context of the COVID-19 pandemic, from a gender perspective.

7. It remains to be clarified whether it was possible to identify the analysis of the aspects that determine health and well-being from a gender perspective in the methods and present it in the conclusions in people with confirmed Covid-19.

R=According to the purpose of the study, we analyzed which aspects of positive coping are linked to the stress response, and with the presence/absence of important metabolic diseases in Mexico and in the context of the COVID-19 pandemic, from a gender perspective. Sadly we do not have access to the data for the COVID confirmed data. 

The selection bias that occurs when selecting people who consult a health service for symptoms of Covid-19 must be declared, since they may have different conditions or characteristics from those who do not consult.

R= The data was collected from those people who requested a diagnostic test for COVID-19. Therefore, the possible deviation could be in the fact that only those who arrived were sampled and not those who, having symptoms that suggested COVID.19, did not request the diagnosis.

Journal Requirements:

R= The format was checked up 

R=The procedures has been described in the Methods section. 

Following all sanitary measures indicated during the COVID-19 pandemic, participants were verbally informed about the objectives of the protocol; once they agreed to participate, a Google Forms link via What’s App, was sent to them, to obtain informed consent via on line, and answer the general data questionnaire and the psychological tests. Personal data was processed in accordance with the policy of the Institute for Transparency, Access to Public Information, Protection of Personal Data. Once registered, a Google Forms link was sent through WhatsApp to answer a file with general data and psychological tests. This work was carried out following the Code of Ethics of the Declaration of Helsinki and approved by the Committee of Ethical Research of the CESS.

"This research has been funded by Dirección General de Investigaciones of Universidad Santiago de Cali under call No. 01-2022."

"This research has been funded by Dirección General de Investigaciones of Universidad Santiago de Cali under call No. 01-2022. The funders had no role in study design, data collection and analysis, decision to publish, or preparation of the manuscript."

R= The text has been modified and removed.

R=The repository files have been included. 

All were incorporated according to the guide for the author. 

S1_fig

S2_fig

S3_fig

S4_fig

S1_table 

S2_table

S3_table

Reviewers' comments:

Reviewer #1: I recommend the authors to mention the following findings related to gender, COVID-19 and mental health in the Introduction and discussion, I will review this paper again.

R= We have reviewed the articles proposed by the reviewer and written brief comments that summarize their vision, as follows:

At the same time, during the COVID 19 pandemic, numerous reports were made that indicated increased levels of anxiety, stress, and a greater psychological impact on women.

---

## [Decision Letter · Decision Letter 1]

29 May 2023

STRESS SYMPTOMS AND POSITIVE COPING DURING CORONAVIRUS DISEASE 2019: THE NEED TO LOOK AT HEALTH FROM A GENDER PERSPECTIVE

PONE-D-22-34111R1

Dear Dr. Romo-González,

We’re pleased to inform you that your manuscript has been judged scientifically suitable for publication and will be formally accepted for publication once it meets all outstanding technical requirements.

Kind regards,

Diana Marcela Walteros Acero, M.D.

Academic Editor

PLOS ONE

Additional Editor Comments (optional):

The comments were reviewed by the authors and resolved appropriately, the comments of the authors as well as the editor were taken into account.

The manuscript presents original research on aspects of mental health in the Covid-19 pandemic, considering it a topic of interest, it contributes to the generation of new knowledge. The authors declare the source of the resources for the research and the manuscript as well as the conflicts of interest"

Reviewers' comments:

Reviewer's Responses to Questions

**Comments to the Author**

1. If the authors have adequately addressed your comments raised in a previous round of review and you feel that this manuscript is now acceptable for publication, you may indicate that here to bypass the “Comments to the Author” section, enter your conflict of interest statement in the “Confidential to Editor” section, and submit your "Accept" recommendation.

Reviewer #1: All comments have been addressed

2. Is the manuscript technically sound, and do the data support the conclusions?

Reviewer #1: Yes

3. Has the statistical analysis been performed appropriately and rigorously? 

Reviewer #1: Yes

4. Have the authors made all data underlying the findings in their manuscript fully available?

Reviewer #1: Yes

5. Is the manuscript presented in an intelligible fashion and written in standard English?

Reviewer #1: Yes

6. Review Comments to the Author

Reviewer #1: I recommend publication for the paper "STRESS SYMPTOMS AND POSITIVE COPING DURING CORONAVIRUS DISEASE

2019: THE NEED TO LOOK AT HEALTH FROM A GENDER PERSPECTIVE"

:"

7. PLOS authors have the option to publish the peer review history of their article (what does this mean?). If published, this will include your full peer review and any attached files.

Reviewer #1: No

---

## [Editor Report · Acceptance letter]

28 Jun 2023

PONE-D-22-34111R1 

Stress symptoms and positive coping during coronavirus disease 2019: the need to look at health from a gender perspective 

Dear Dr. Romo-González:

I'm pleased to inform you that your manuscript has been deemed suitable for publication in PLOS ONE. Congratulations! Your manuscript is now with our production department. 

Kind regards, 

on behalf of

Dr. Diana Marcela Walteros Acero 

Academic Editor

PLOS ONE